# Self-supervised Monocular Trained Depth Estimation using Self-attention and Discrete Disparity Volume - ML Reproducibility Challenge 2020

## 1         Reproducibility Summary

*Depth estimation is one of the widely studied applications in the field of computer vision as it captures the perception of the 3D world. This knowledge can be used in applications like autonomous vehicles and obstacle warning systems. The paper [1] proposes to estimate the depth by a self-supervised technique where the model is trained using a sequence of monocular input images.*

**Scope of Reproducibility**

The state-of-the-art results in depth estimation are mostly based on fully-supervised techniques. Although recent works in unsupervised techniques have shown promising results, the performance gap is still prominent and they mostly rely on stronger supervision signals (stereo-supervision or monocular+stereo supervision). The paper proposes to close this performance gap with the fully-supervised methods using only the monocular sequence for training with the help of additional layers - self-attention and discrete disparity volume.

**Methodology**

The architecture proposed in the paper was implemented from scratch since the code is not available online. Ablation study, additional experiments using hyperparameters were performed and the results obtained from reproduction were compared with that claimed in the paper. The experiments were trained and tested using GeForce GTX 1080 GPU.

**Results**

We reproduced the performance proposed in the paper to within 2% of the reported values. Therefore, it can be inferred that the use of self-attention and discrete disparity volume helped in improving the performance using only the monocular sequence of images.

**What was easy**

Reproduction of the encoder in the proposed algorithm was easily handled. The description of the self-attention layer is elaborate and could be conveniently reproduced.

**What was difficult**

The unavailability of the code online made it inevitable for the reproduction of the algorithm from scratch. The process of training using the ResNet101 encoder was time-consuming, however, the memory utilization was reduced to half due to the usage of activated batch normalization.

**Communication with original authors**

The authors helped in understanding concepts like the usage of dilated convolution and discrete disparity volume. This played a crucial role in reproducing the results reported in the paper.

# 1  Introduction

Monocular depth estimation is a technique of inferring the depth of objects from a single sequence of input images. It is often referred to as an ill-posed problem as more than one point in 3D converges to the same point in 2D. Traditional fully supervised learning methods [2] [3] that have achieved state-of-the-art results in the field of depth estimation, use ground truth data obtained from LIDARs to estimate the depth map. Acquiring ground truth data is a tedious task since they require accurate instrumentation and synchronization of data streams from multiple sensors. Therefore, unsupervised techniques are preferred. In the unsupervised technique, we have two kinds - stereo learning and monocular learning. Monocular learning uses simple readily available monocular cameras hence are preferred over stereo learning which uses an expensive stereo camera. In monocular technique, the depth information is extracted from the sequence of images separated across time. Recent works [4] [5] on monocular depth estimation use dual networks - disparity and pose - to achieve promising results. However, they mostly rely on stronger supervision signals(stereo-supervision or monocular+stereo supervision). This paper "Self-supervised Monocular Trained Depth Estimation using Self-attention and Discrete Disparity Volume" [1], improves on the existing work by using only monocular sequence to train and close the performance gap between self-supervised and fully-supervised models.

## 1.1  Background

Depth estimation is crucial to understand the 3D world for navigation assistance in computer vision and robotics systems. Several methods to estimate depth have been adopted over the years. Initially, active sensors like LiDAR and RADAR were used on the navigating agent [6]. However, due to its limited range, increased noise, power consumption, and cost, a deep convolutional neural network with passive sensors became popular. Deep convolutional neural network methods can be mainly categorized into 3 major types - 1. Fully supervised learning, 2. Self-supervised stereo learning, and 3. Self-supervised monocular learning.

Fully supervised methods require input images along with their corresponding depth maps for the network to train. Obtaining a huge amount of ground truth depth data in all the environments where the navigating agent is deployed can be tedious. However, the performance of this method holds the current state-of-the-art values for depth estimation. Next, we have self-supervised stereo learning which does not require ground truth data but instead relies on the left and right view of the camera. While self-supervised monocular learning is considered a good alternative because of its ease in collecting the input data, it poses many problems. Depth estimation using monocular sequence is inherently ambiguous and an ill-posed problem. Recent advances leverage monocular images to estimate depth by adopting a dual network. One such architecturally advanced solution is proposed by this paper [1] to close the performance gap with fully supervised methods.

## 1.2  Summary of the Paper

This research [1] aims to outstrip the existing state-of-the-art depth estimation algorithm in terms of performance. To achieve this, two techniques are introduced. The Self-attention layer is used to extract the global context of an image which contrasts the traditionally used convolution layer that captures only the local context. And the other technique used is the multi-scale discrete disparity volume decoder. This technique is mainly used to obtain more robust and sharper depth maps from the network. Discrete disparity volume helps in estimating pixel-wise uncertainty, which is crucial for refining depth estimation.

The paper adopts the pose estimating neural network and loss functions - photo-metric re-projection loss and edge-aware smoothness loss - from Monodepth2 [5]. The depth network is used to predict the depth map $D_t$ of image $I_t$. While the pose network is used to predict the transformation matrix having 6 degrees of freedom using image frames $I_t$ and $I_{t`} \in \{I_{t-1}, I_{t+1}\}$ where $I_{t`}$ are the temporally adjacent frames to $I_t$. Images $I_{t`}$ are transformed to $I_t$ using the transformation matrix resulting in $I_{t` \to t}$ and the photo-metric re-projection loss is given by the L1 distance in pixel space between $I_t$ and $I_{t` \to t}$. Edge-aware smoothness loss helps to predict smooth disparity estimates in local neighborhoods and penalizes drastic depth changes in flat regions. The total loss is equivalent to the sum of photo-metric re-projection and weighted edge-aware regularization. And the auto-masking technique is used to preserve the static scene-moving camera assumption.

The rest of the paper is organized as follows. Section 2 gives an overview of the claims made in [1]. Section 3 elaborates on the methodology, computational and experimental requirements used for reproducing the proposed algorithm. Section 4 contains the reproduced results claimed in the paper along with additional experimental results. In Section 5 we have the analysis of the reproduction.

## 2   Scope of reproducibility

The paper [1] proposes to achieve quality performance using only the monocular sequence for training using two techniques namely self-attention and discrete disparity volume. The following are the major claims made in the paper -

- Addition of discrete disparity volume and self-attention layers show steady improvement in all evaluation measures.
- The performance obtained from the proposed architecture closes the gap between self-supervised and fully-supervised models.

## 3   Methodology

The goal of this study is to validate the claims made by the original paper and extend the analysis to other datasets and methodologies. First, we implemented the approach with the help of the description provided in the paper, as the original code is unavailable online. The implementation of the self-attention layer in [7] and discrete disparity layer in [3] were useful for the reproduction.

### 3.1   Model descriptions

The baseline depth architecture is adopted from [5]. The depth architecture has two main parts – encoder and decoder. The image sequence is fed as input to the ResNet101 encoder with atrous spatial pyramid pooling (also known as a dilated convolution in [1]), which learns the features of the input image to generate an output that is 1/8th the size of the input image. ResNet101 use pre-trained ImageNet weights. The encoder is adjoined with a self-attention layer. This layer explores the global context of the input image, unlike convolution layers which extract only the local context of the image. A convolution layer is used to reduce the channels from the attention layer to 128, which is the number of bins used by discrete disparity volume. The decoder comprises a multi-scale discrete disparity volume layer to obtain a robust and sharper depth estimate. Additionally to reduce the exhaustive memory usage of ResNet101, activated batch normalization [8] which fuses batch normalization with activation function is used to bring the memory usage down by up to 50%. The loss functions used and the pose network is the same as the Monodepth2 [5] implementation.

### 3.2   Datasets

KITTI Raw [9] and Cityscapes [10] datasets were used in the experiments. The architecture is trained and evaluated using the KITTI dataset. The train and validation set are split using the traditional Eigen et al. [2] method, which gives 39,810 monocular training sequences and 4,424 validation sequences. Training monocular images are reduced to 640 x 192 pixels and the following data augmentations are performed with 50% probability - horizontal flip, random contrast($\pm0.2$), saturation($\pm0.2$), hue jitter($\pm0.1$), and brightness($\pm0.2$). The depth for this dataset is evaluated up to a fixed range of 80m. This value is the same as that used in Monodepth2 and the original authors of paper [1]. Corresponding results are recorded in this reproducibility.

The method was quantitatively compared with the held-out test set of Cityscapes datasets.

### 3.3   Hyperparameters

The pose and depth network is jointly optimized using Adam Optimizer [11] with $\beta_1 = 0.9$, $\beta_2 = 0.999$ and learning rate of $1e^{-4}$. The training process is executed for 20 epochs and a single learning rate decay of $1e^{-5}$ is applied after 15 epochs. And the final loss is computed by setting the smoothness term $\lambda = 1e^{-3}$.

### 3.4   Experimental setup and code

Monodepth2 [5] implementation was used as reference to perform ablation experiments. The same environmental setup - python=3.6.6, torch=0.4.1 and torchvision=0.2.1 - as Monodepth2 was used. Initially, the self-attention layer was included in the encoder. This was followed by replacing the decoder with multi-scale discrete disparity volume. Finally, ResNet101 with atrous spatial pyramid pooling was used as an encoder in -place on ResNet18.

In the final model proposed, the encoder is adopted from the OCNet [7] architecture. ResNet101 is used with minor modifications by replacing the convolutions in the last two blocks with dilated convolutions of rate 2 and 4 respectively. This is then fed to the self-attention layer. The atrous spatial pyramid pooling module is the result of the concatenation of the output of self-attention layer along with four other dilated convolution layers of rates [1, 12, 24, 36]. This

architecture of the encoder is not easily deduced from the overall architecture(Figure 2 in [1]) proposed in the paper. Referring to OCNet [7] helps in better understanding. The decoder adopts the concept of multi-channel ordinal regression from DORN [3]. The proposed architecture makes use of uniform discretization. For this reproduction, the code snippet for the same was provided by the authors of [1].

Evaluation metrics like absolute relative, square relative, and RMSE log loss measures are used. The lower these metric values are the better it is. And accuracy is also calculated with three different delta values. An increase in accuracy suggests better performance.

The code and references for each of the layers mentioned above can be found at https://github.com/sjsu-smart-lab/Self-supervised-Monocular-Trained-Depth-Estimation-using-Self-attention-and-Discrete-Disparity-Volum

## 3.5 Computational requirements

The proposed algorithm was implemented using Pytorch [12] framework and the experiments were trained and tested using NVIDIA Tesla P100 GPU and CPU environment - Intel Xeon E5-2660 v4 (2.0GHz, 35M Cache). The ResNet18 models were executed with a batch size of 8 while the ResNet101 models were executed with a batch size of 4. Table 1 shows the computational requirements like memory usage, training time and inference time on the KITTI Raw dataset [9]. We observe that as each additional layer is added the resource utilization increases substantially.

| Backbone | Self-Attn | DDV | Training time (hr) | Inference time (ms) | | Memory (MB) |
| --- | --- | --- | --- | --- | --- | --- |
| | | | | GPU | CPU | |
| ResNet18 (Baseline) | ✗ | ✗ | 12 | 6.36 +/- 0.69 | 105.41 +/- 4.23 | 46.8 |
| ResNet18 | ✗ | ✓ | 111 | 359.20 +/- 3.13 | 952.9 +/- 16.17 | 52.8 |
| ResNet18 | ✓ | ✗ | 31 | 6.35 +/- 3.13 | 1020.07 +/- 245.29 | 64.2 |
| ResNet18 | ✓ | ✓ | 102 | 361.02 +/- 2.45 | 947.74 +/- 12.95 | 52.8 |
| ResNet101 w/ ASPP | ✗ | ✗ | 84 | 16.95 +/- 1.57 | 8039.17 +/- 38.18 | 464.8 |
| ResNet101 w/ ASPP | ✗ | ✓ | 205 | 668.24 +/- 2.29 | 6145.43 +/- 19.74 | 258.4 |
| ResNet101 w/ ASPP | ✓ | ✗ | 89 | 18.23 +/- 1.40 | 8045.26 +/- 18.29 | 464.8 |
| ResNet101 w/ ASPP | ✓ | ✓ | 204 | 656.23 +/- 0.98 | 6108.50 +/- 12.23 | 252.7 |

Table 1: **Qualitative Analysis.** The training and inference execution time for different versions of the model using KITTI Raw test data is listed. Memory utilization by each of these models are also given. Monodepth2 [5] is considered as the baseline architecture.

The GPU inference time and training time with the addition of only the self-attention layer is less when compared with the addition of only discrete disparity volume. However, the CPU inference time and the memory utilized with the addition of only self-attention is higher when compared with the addition of only discrete disparity volume to the architecture. The reduced memory utilization in the architecture with both self-attention and discrete disparity volume layers is credited to the use of activated batch normalization.

## 4 State-of-the-art Results

The results were obtained after implementing with the help of the description given in the paper. The evaluation metrics of the final reproduced model are close to the values reported in the paper. A subset of the attention maps obtained from the proposed methods is shown in Figure 1.

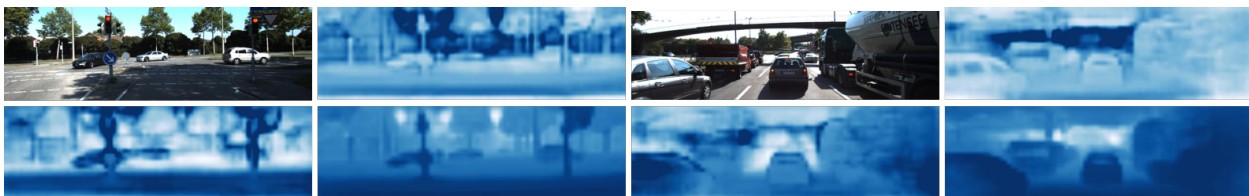

Figure 1: **Attention maps.** Subset of the attention maps produced by the proposed method. Blue indicates the region of attention.

### 4.1 Results reproducing original paper

The following are the results obtained to validate the claims(Section 2) made in the paper. All the experiments on training were performed on the KITTI Raw dataset [9].

#### 4.1.1 Ablation Study

The performance after introducing each new layer - self-attention, discrete disparity volume and ResNet101 with atrous spatial pyramid pooling was recorded as part of the ablation study.

| Backbone | Self-Attn | DDV | Abs Rel | Sq Rel | RMSE | RMSE log | $\delta < 1.25$ | $\delta < 1.25^2$ | $\delta < 1.25^3$ |
|---|---|---|---|---|---|---|---|---|---|
| ResNet18 (Baseline) | ✗ | ✗ | 0.115 | 0.903 | 4.863 | 0.193 | 0.877 | 0.959 | 0.981 |
| ResNet18 | ✗ | ✓ | 0.118 | 0.941 | 4.964 | 0.195 | 0.869 | 0.957 | 0.981 |
| ResNet18 | ✓ | ✗ | 0.115 | 0.970 | 4.829 | 0.192 | 0.880 | 0.959 | 0.981 |
| ResNet18 | ✓ | ✓ | 0.116 | 0.900 | 4.909 | 0.194 | 0.871 | 0.957 | 0.981 |
| ResNet101 w/ ASPP | ✗ | ✗ | 0.109 | 0.848 | 4.712 | 0.188 | 0.885 | 0.961 | 0.982 |
| ResNet101 w/ ASPP | ✗ | ✓ | 0.109 | 0.871 | 4.738 | 0.188 | 0.886 | 0.961 | 0.981 |
| ResNet101 w/ ASPP | ✓ | ✗ | 0.108 | 0.864 | 4.677 | 0.186 | 0.887 | 0.962 | 0.982 |
| ResNet101 w/ ASPP | ✓ | ✓ | 0.108 | 0.831 | 4.694 | 0.186 | 0.885 | 0.961 | 0.982 |
| Paper [1] | ✓ | ✓ | 0.106 | 0.861 | 4.699 | 0.185 | 0.889 | 0.962 | 0.982 |

Table 2: **Ablation Study.** Results for different versions of the model with comparison to the baseline Monodepth2 [5] using KITTI Raw test dataset. Metrics indicated by red: *lower is better*, Metrics indicated by blue: *higher is better*

From the overall reproduced model in Table 2, it can be inferred that the absolute relative loss is around 2% behind the value stated in the paper while the square relative loss value is improved by around 3%. The other metric values are close to the stated values. Therefore, it can be concluded that the overall performance of the reproduction is almost in equivalence with the results proposed in the paper [1].

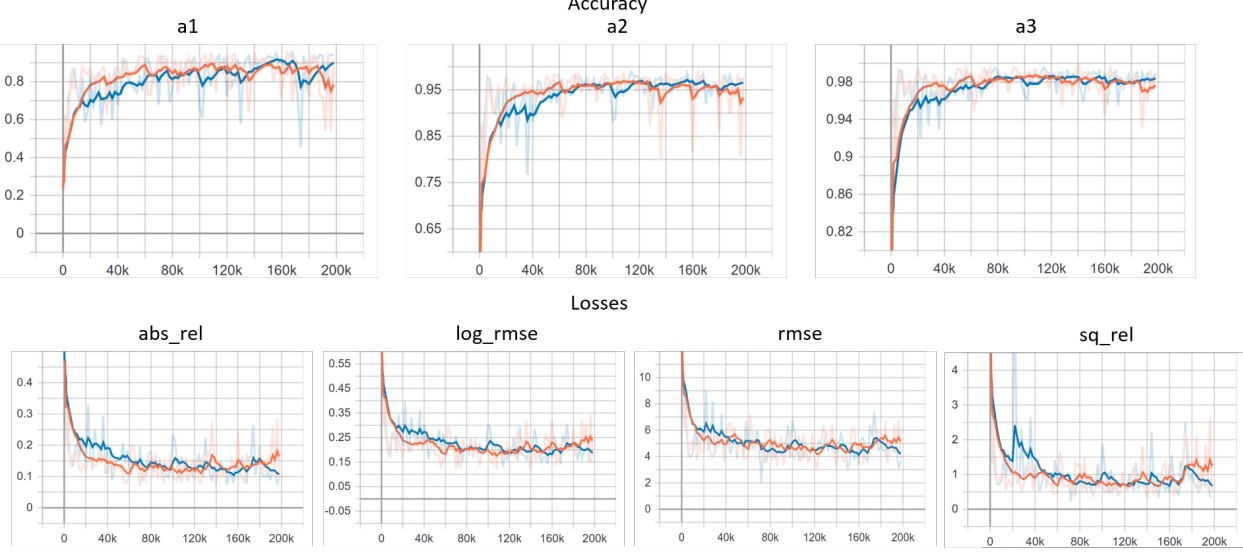

Figure 2: Accuracy and loss curves of training and validation data through the process of training the final model. Orange represents the train set trend and blue represents the validation set trend

#### 4.1.2 Quantitative Analysis of the Performance

The current state-of-the-art result in the field of depth estimation is held by a fully-supervised method [3]. However, self-supervised monocular learning being more convenient has shown considerable performance gain. Monodepth2 [5], which previously held the highest state-of-the-art values for self-supervised monocular learning was about 60% behind DORN [3]. The performance of the reproduced model surpasses Monodepth2 [5], and is about 47% behind [3]. Thus, it can be inferred that the use of ResNet101 with atrous spatial pyramid pooling, self-attention, and discrete disparity

volume helped in closing the performance gap with fully-supervised methods using only the monocular sequence of images for training.

| Method | Train | Abs Rel | Sq Rel | RMSE | RMSE log | $\delta < 1.25$ | $\delta < 1.25^2$ | $\delta < 1.25^3$ |
|---|---|---|---|---|---|---|---|---|
| Monodepth2 [5] | M | 0.115 | 0.903 | 4.863 | 0.193 | 0.877 | 0.959 | 0.981 |
| Paper [1] | M | 0.108 | 0.833 | 4.682 | 0.186 | 0.887 | 0.962 | 0.982 |
| DORN [3] | D | 0.072 | 0.307 | 2.727 | 0.120 | 0.932 | 0.984 | 0.994 |

Table 3: **Quantitative Analysis.** Results for different methods of depth estimation on KITTI Raw test dataset. Metrics indicated by red: *lower is better*, Metrics indicated by blue: *higher is better*. D - Depth supervision, M - Self-supervised monocular training

### 4.1.3 Identifying sensitive regions

The paper proposes to achieve sharper results on sensitive regions like thinner structures such as poles and reflective surfaces. From the Figure 3 this can validated.

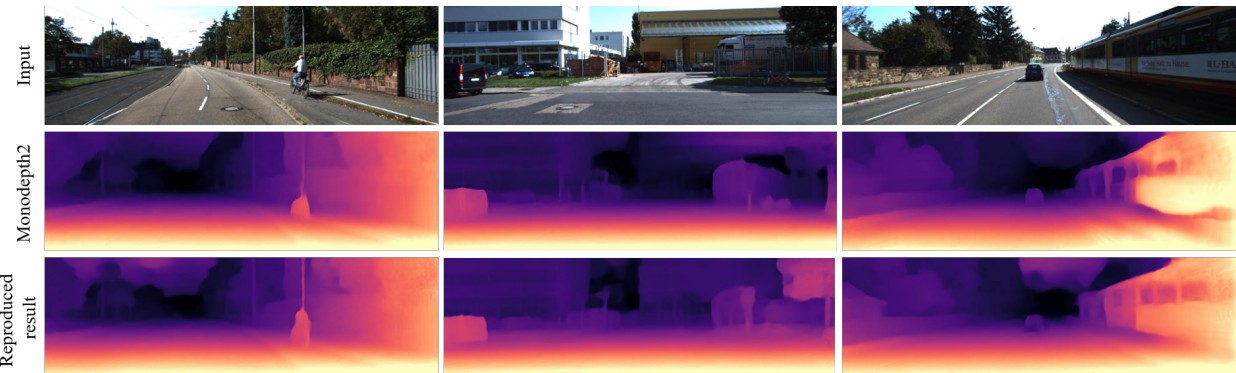

Figure 3: Comparison of Monodepth2 [5] and reproduced model [1] on KITTI test set. Left: shows the model reproduced from the paper predicts the depth of overlapped objects(cycle and pole) better than the Monodepth2 model. Middle: Thinner structures like the signboard and tree trunk are well represented with the reproduced model. Right: The reproduced model succeeds in estimating the depth of reflective surfaces like the train windows better than Monodepth2

## 4.2 Results beyond original paper

The following additional experiments were conducted to further validate the results proposed in the paper.

### 4.2.1 Inference on Cityscapes dataset

The model obtained from training the proposed algorithm on KITTI Raw dataset [9] is tested on the held out test set of the Cityscapes [10] data to analyze the generalizability of the trained model. The result of this experiment is recorded below (Table 4).

| Method | Abs Rel | Sq Rel | RMSE | RMSE log | $\delta < 1.25$ | $\delta < 1.25^2$ | $\delta < 1.25^3$ |
|---|---|---|---|---|---|---|---|
| Monodepth2 [5] | 0.242 | 3.018 | 11.300 | 0.332 | 0.572 | 0.844 | 0.940 |
| Paper [1] | 0.232 | 2.870 | 11.206 | 0.326 | 0.587 | 0.852 | 0.943 |

Table 4: **Qualitative Analysis.** Performance obtained using Cityscapes test set. Metrics indicated by red: *lower is better*, Metrics indicated by blue: *higher is better*.

From Table 4, it can be observed that like with the KITTI Raw dataset, the model reproduced from [1] perform better than Monodepth2 [5] when tested using the Cityscapes dataset as well.

### 4.2.2 Comparison between Discretization Methods

The decoder network is adopted from the research proposed in DORN [3] where ordinal regression is used to predict depth. DORN proposes two types of discretization methods that can be used as part of the decoder i.e. uniform discretization (UD) and space increasing discretization (SID). The following experiment is to compare the performance between these two discretization methods.

| Discretization | Abs Rel | Sq Rel | RMSE | RMSE log | $\delta < 1.25$ | $\delta < 1.25^2$ | $\delta < 1.25^3$ |
|---|---|---|---|---|---|---|---|
| UD | 0.108 | 0.833 | 4.682 | 0.186 | 0.887 | 0.962 | 0.982 |
| SID | 0.108 | 0.850 | 4.689 | 0.186 | 0.887 | 0.961 | 0.982 |

Table 5: Comparison of performance obtained using different discretization methods on KITTI Raw test set. Metrics indicated by red: *lower is better*, Metrics indicated by blue: *higher is better*.

| Discretization | Training time (hr) | Inference time (ms) | | Memory (MB) |
|---|---|---|---|---|
| | | GPU | CPU | |
| UD | 199 | 656.23 +/- 0.98 | 6108.50 +/- 12.23 | 252.7 |
| SID | 207 | 669.75 +/- 1.05 | 6186.62 +/- 29.24 | 258.4 |

Table 6: Comparison of resource utilization obtained using different discretization methods.

From Tables 5 and 6 it is observed that as mentioned in [3] there is no significant difference in the performance between both the discretization methods. However, the training time, resource utilized and inference time of SID is slightly more when compared with that utilized by UD. Most importantly UD is a single-line implementation using torch.linspace function while SID is not. Therefore the authors of [1] prefer UD over SID.

## 5   Discussion

Self-supervised monocular learning is widely appreciated for its ease in collecting input training data. However, its performance falls back when compared with stereo learning and fully-supervised learning. We would acknowledge this paper for its novel contribution to close this performance gap through techniques like self-attention and discrete disparity volume.

### 5.1   Discrete Disparity Volume

The decoder of the architecture which constitutes multi-scale discrete disparity volume could be elaborated for better understanding. Discrete disparity volume is a strategy through which the continuous depth value is divided into discrete values [3]. This process is referred to in [3] as ordinal regression. Unlike a regular regression process where the depth value at each pixel is predicted from the continuous range of values, ordinal regression is more like a multiclass classification process where the depth values are predicted from the collection of discrete values. The discretization process can be done using either space-increasing discretization(SID) or uniform discretization(UD). In UD the range of continuous depth values $[\alpha, \beta]$ are equally divided into bins with discrete threshold values given by equation 1

$$t_i = \alpha + (\beta - \alpha) * i/K, \tag{1}$$

However, since the error maximizes at higher depth values, UD tends to strengthen the loss at these large depth values. Therefore, SID is introduced to down weight the larger depth values. Discrete threshold values of SID is given by the equation 2

$$t_i = e^{\log(\alpha) + \frac{\log(\beta/\alpha) * i}{K}}, \tag{2}$$

where $t_i \in \{t_0, t_1, ..., t_K\}$ are discretization thresholds and K is the number bins needed (K=128, as given in paper). Therefore, the research in [3] suggests that SID outperforms UD when used as a decoder in depth network. However, the paper [1] proposes a decoder that makes use of UD to generate its performance. Unlike in DORN [3], from the experiments we conducted during the process of reproduction, it is observed that there is no significant difference in performance between UD and SID. We believe that this difference between results of DORN [3] and the paper [1] merits for more research.

## 5.2 Observations

The paper proposes an advanced architectural enhancement to close the performance gap with fully-supervised methods. Depth estimation is crucial to understand the surrounding environment. Therefore, it is one of the important pieces of information needed for autonomous navigation systems. The improved precision obtained in this paper is beneficial for autonomous navigation applications to be commercially viable. However, this complex architecture proposed occupies extensive computational resources and it also has an extended response time as shown in Table 1. The performance improvement when qualitatively analyzed is better than the previous state-of-the-art by a few decimals. When an application in hand is more towards assisting humans rather than making decisions, then a method with a simpler architecture can be preferred over the architecture proposed in the paper.

The previous state-of-the-result was held by Monodepth2 [5] This architecture comprises of ResNet18 encoder, depth decoder, and the pose network. It occupies 81% smaller memory when compared with the reproduced architecture. But the performance of the reproduced model is higher than Monodepth2. Therefore, appropriate models that fulfill the requirements of the application can be used.

For future work, we would like to repeat the experimentation with increased batch size and vary hyper-parameters like weight decay to study the variations in performance. Also, we would explore techniques to reduce the computational footprint and inference time of this complex architecture.

## 5.3 What was easy

The OCNet architecture [7] referenced by the authors could be adopted conveniently to implement the encoder architecture. It enfolds ResNet101, atrous spatial pyramid pooling, and self-attention layers. Ablation experiments with ResNet18 could be performed faster as it is smaller when compared to ResNet101.

## 5.4 What was difficult

The unavailability of the code online made it inevitable for the implementation of the algorithm from scratch. The process of training using the ResNet101 encoder was time-consuming, however, the memory utilization was reduced to half due to the usage of activated batch normalization. The following are the concepts that were not straightforward in the paper and required the author's input to understand and implement -

- The implementation of the encoder with atrous spatial pyramid pooling cannot be perceived using Figure 2 of original paper [1]. Reference to OCNet architecture as prompted by the authors helped in encoder implementation.

- The description of discrete disparity volume was limited. Reference to DORN [3] and inputs from authors about the UD usage helped implement the decoder.

## 5.5 Communication with original authors

Warm regards to the authors, Adrian Johnston and Gustavo Carneiro for guiding us through the process. They helped clarify the experiment settings and guided us with the right resources to accurately reproduce the experiments under the same settings. They pointed us to the OCNet architecture [7], which reduced the workload. Also, they answered our questions on discrete disparity volume and also provided the code snippet for the same.

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
