# OpenReview forum: "Self-supervised Monocular Trained Depth Estimation using Self-attention and Discrete Disparity Volume - ML Reproducibility Challenge 2020"
_ML_Reproducibility_Challenge/2020 — Reject_

### Official Review · AnonReviewer3 · 2021-02-20
**Reproducibility results on Make3D and pixel-wise uncertainty missing; unexplained difference in the training parameters**

**Rating:** 4
**Confidence:** 5

**Review:**

The authors provide a nice summary of the original article in the first section of the paper. Chapter 5.1. on Discrete Disparity Volume is a nice addition for further clarification. However, sometimes I believe they could have been more clear on a number of occasions, especially when it comes to the experimental results. Here are some comments that should be addressed:

- The authors of the reproducibility study coded everything from the scratch. For the purpose of the clarity they should have commented on the architecture in Figure 2 of the original paper: was everything clear and straightforward to implement? where there any ambiguities or points where a decision was to be made that could not be deduced from the Figure 2? For example, a bit more details on OCNet incorporation could be useful for the readers who would like to get a complete picture of the reproduction. This is also a part where they got help from the authors of the original article.
- In Section 3.2 on line 110 it is reported that the depth up to 100m were used for training on KITTI dataset. However, in the original article,  the authors report using depths up to 80m. Why the discrepancy and have they asked the authors of the original paper why did they decide on 80m instead? What would be the reproduced results if the reproducibility authors have used 80m as well? The results they report in the reproducibility manuscript are compared against the results from the original article,  yet there seems to be a difference in the training data. **This is not a good practice.**
- In all results reported in Chapter 4 there is always a piece of information not being explicitly stated. For example, in Section 4.1.1 (Table 2) and Section 4.1.2 (Table 3), one needs to read up to the beginning of Chapter 4 to conclude that these results are for KITTI dataset. In Section 4.2.2 (Table 5), one needs to compare results from Table 4 (Cityscape) and Table 3 (KITTI) to conclude that KITTI was used in Table 5 results. Furthermore, in Section 4.2.2 it could also be more clear what bases were used
- Furthermore, there are no results reported on the Made3D dataset, although this was explicitly stated by the reproducibility authors in Section 3.2. on line 111.
- There are no results reported for attention maps from the self-attention module.
- There are no results reported for pixel-wise depth uncertainty.
- Chapter 2, line 87: this is not a major claim or hypothesis of the original paper. It is more of a side-result that can be easily visually verified (check 2.2 and 2.3 of the original paper).
- Tables 1 and 2: explicitly mentioning what the baseline is (Monodepth2 with ResNet18), would help the readers.
- In Table 3: I believe Monodepth2 should be classified as "M" (self-supervised), while DORN should be "D" (supervised).
- Table 6: The reported increase in memory consumption is not as dramatic as the authors claim. Could it be that one is simpler than the other? Have they tried asking the authors of the original paper what they used for the results reported in the original paper?

Some minor comments:

- There is a few minor grammatical errors. A bigger issue is that sometimes the wording and the sentence structure is a bit harder to follow and understand. This could be improved.
- I am not sure why the reproducibility authors prefer to use the term "atrous spatial pyramid pooling" instead of "dilated convolution" (as used in reporting the results in the original paper), but they should (gently) mention that these two terms refer to the same concept.
- The authors wrote code cleanly and it is not hard to read. On a few places a few more comments would make things easier to read, but it is not a big issue.

The authors did a good coding job, but the paper itself could have been better written. There is still work to be done to make it more clear and self-explanatory when it comes to the implementation details and the reported results.

**Familiar With The Original Paper:**

I have read the original paper

**Reproducibility Summary:**

Report has summary

---

### Official Review · AnonReviewer1 · 2021-03-03
**Good report but need more implementation details**

**Rating:** 6
**Confidence:** 4

**Review:**

The report aims to reproduce and evaluate the paper [1] . The paper's main claims are that it produces nearly SOTA monocular depth estimation results through a combination of techniques - self attention and discrete disparity volume. To this end, the report attempts to reconstruct these results, borrowing from other sources and code available (e.g. Monodepth2 [2]).

Clarity: The report is well written and easy to read.
Originality: The report and code seems to be the first attempt at reconstructing/implementing the paper.
Significance: As monocular depth estimation is an important computer vision task in scenarios such as autonomous driving, the work is significant.

Pros and cons:

Pros: With regards to pros, generally, the results presented align with the paper's claims that self-attention and DDVs help in improving monocular depth estimation. The results beat Monodepth2 convincingly, but cannot match DORN [3].

Cons: The paper does not explain the underlying concepts properly. I would have hoped for some explanations on self-attention implementation and in particular, the discrete disparity volume ideas. Examples:

1) How is the DDV constructed and what computational/implementation challenges did it pose?
2) How was the system tuned?
3) Comments on the Atrouss Spatial Pyramidal Pooling?
4) A system diagram with explanation and some implementation notes on components


[1] http://openaccess.thecvf.com/content_CVPR_2020/papers/Johnston_Self-Supervised_Monocular_Trained_Depth_Estimation_Using_Self-Attention_and_Discrete_Disparity_CVPR_2020_paper.pdf

[2] Monodepth2: https://arxiv.org/pdf/1806.01260.pdf

[3] DORN: https://arxiv.org/pdf/1806.02446.pdf

**Familiar With The Original Paper:**

I have read the original paper

**Reproducibility Summary:**

Report has summary

---

### Decision · Program_Chairs · 2021-03-31

**Decision:**

Reject

**Comment:**

Overall reviews and/or the paper content not good enough for the AC to recommend to the journal.